# Structural Characteristics and Microstructure Analysis of Soft Soil Stabilised with Fine Ground Tile Waste

**DOI:** 10.3390/ma16155261

**Published:** 2023-07-26

**Authors:** Mohd Hafizan Md Isa, Suhana Koting, Huzaifa Hashim, Salsabila Ab Aziz, Syakirah Afiza Mohammed

**Affiliations:** 1Center for Transportation Research, Department of Civil Engineering, Faculty of Engineering, University of Malaya, Kuala Lumpur 50603, Malaysia; hafizan_isa@um.edu.my (M.H.M.I.); huzaifahashim@um.edu.my (H.H.); salsabila@jkr.gov.my (S.A.A.); 2Faculty of Civil Engineering & Technology, Universiti Malaysia Perlis, Perlis 02600, Malaysia

**Keywords:** tile waste, soil stabilisation, unconfined compressive strength, flexural strength, indirect tensile strength, microstructure analysis

## Abstract

Using ceramic tile waste as a soil stabiliser in road construction is a potential solution to dispose of the waste material while providing a cost-effective alternative to traditional stabilising agents. The ceramic tile waste, when crushed and mixed with soil, helps to improve the strength and durability of the road base. However, the effectiveness of the ceramic tile waste as a soil stabiliser depends on the type and size of ceramic tiles used and the soil properties being stabilised. This study investigated the effect of ground tile waste on the plasticity, compatibility, and mechanical properties such as the unconfined compressive strength (UCS), indirect tensile test (IDT), flexural test (FS), and microstructural analysis. A range of soil mixtures was prepared by adding the different percentages of fine tile waste (TW): 5% to 40%. Including tile waste in the soil led to a decrease in its water-holding capacity, reducing the optimum moisture content required for optimal compaction. Meanwhile, the maximum dry density increased. The UCS, IDT, and FS improved when the optimum 15% of TW was used in the mixes. However, the strength decreased after 20% of the TW addition. This effect was particularly pronounced in the presence of excessive TW contents in soil samples without a pozzolanic reaction. Reusing tile waste as a soil stabiliser can significantly reduce the costs of purchasing new materials and helps to conserve natural resources and reduce the environmental impact of waste disposal.

## 1. Introduction

Soft soil is generally highly compressible and has low shear strength. These issues often lead to embankments with a low bearing capacity that tend to settle excessively [1]. Infrastructure like roads, sewers, pipelines, and manholes constructed on prepared platforms over soft soil will settle over time, as the soft soil requires a long settling period after being filled [2]. The loads transmitted through the pavement’s surface will settle and compromise the stability of the subgrade. The subgrade must have sufficient strength and stiffness to distribute the loads evenly and prevent excessive deformation or failure. Proper evaluation of the subgrade’s mechanical properties and good pavement structure design can help ensure the subgrade’s integrity and prolong the pavement’s service life.

One of the geotechnical design criteria that must be considered in site selection is the bearing capacity of the subgrade. Poor subgrade quality can be addressed with a viable solution, excavation, and substitute with a high-quality backfill [3]. Soil stabilisation is one of the techniques generally employed to enhance the geotechnical and physical characteristics of soil [4]. The typical soil stabilisation technique used in road construction is adding a binder to the subgrade. Some of the most common binders that can improve the strength and stability of the subgrade are cement, lime, fly ash, and construction and demolition waste [5,6,7,8,9]. Each agent binder has different properties that can positively affect the road construction of the subgrade layer. The primary purpose of using a stabilising binder is to improve the shear strength and stiffness of the subgrade and to increase the overall performance of the road structure.

In recent years, public interest in general waste management issues has increased, especially in industrial waste and waste from the construction sector [10]. The increasing awareness of the negative impact of waste on the environment and human health has led to heightened public interest in waste management practices [11]. The construction sector, in particular, generates a significant amount of waste. Construction waste is any material or substance generated during a building or structure’s construction, renovation, or demolition [12]. This waste can come from various sources, including the loss of construction materials at construction sites [13].

The production process for ceramic materials often generates a considerable volume of waste, with approximately 15–30% of the manufactured materials being discarded [14]. The construction industry (construction and demolition) generates the most waste worldwide. Residential and commercial construction projects are the primary source of ceramic waste [15]. The utilisation of waste as a construction material or for soil stabilisation is a rapidly evolving field within the construction industry [16]. Ceramic tiles are known for their robust, durable, and beautiful surfaces. However, due to the popularity of ceramic tiles in recent years, the number of tiles produced has also increased. Cracked, chipped, or damaged tiles are no longer usable for construction and can end up in waste [17]. Despite the potential for recycling ceramic tile waste, the traditional method of disposing of these tiles is still predominantly through landfilling. This is due to a need for standardised recycling practices, the limited availability of recycling facilities, and insufficient expertise and awareness regarding the reuse of ceramic waste [18]. The waste produced causes environmental inconvenience and risks [19]. This has led to a significant problem in disposal as it is impossible to burn or bury it [20]. Reusing tile waste to stabilise the subsoil can minimise the negative impact of this waste on the environment.

Recycling tile waste reduces the need for extracting raw materials and conserves natural resources. It also reduces the amount of waste sent to landfills and lowers greenhouse gas emissions. Mohammed et al. [21] remarked that using waste materials to replace original materials for road construction will lessen the negative effect of the waste materials on the environment. The reuse of solid wastes is gaining importance for sustainable waste management and is being used as standalone stabilisers or as additives to enhance the performance of conventional stabilisers [22]. This leads to a green and sustainable building by not further exploiting the natural resources, apart from exhibiting the effective management of solid waste. Effective waste management is important for reducing the environmental impact, conserving natural resources, reducing costs, and promoting sustainability. Several nations have implemented regulations and policies to improve the management of construction and industrial waste and encourage adopting sustainable waste management practices [23].

Many researchers have conducted extensive application studies on tile waste in soil stabilisation. Studies have shown the potential of tile waste to improve the engineering properties of various soil types. For instance, Onakunle et al. [24] demonstrated the effectiveness of ceramic waste dust derived from tile waste in stabilising lateritic soil in Agbara, Nigeria. Incorporating ceramic waste dust improved soil strength and stability, providing an environmentally friendly and cost-effective solution. In addition, using tile waste has successfully stabilised marine clay and soft clay soils. Al-Bared et al. [25] investigated the sustainable improvement of marine clay using recycled blended tiles, showing promising results in enhancing the geotechnical properties of the soil. Similarly, Al-Bared et al. [26] studied the sustainable strength improvement of soft clay stabilised with recycled additives, highlighting the potential of tile waste to enhance the stability and strength of soft clay soils.

Furthermore, tile waste has effectively stabilised expansive soils known for their high shrink–swell potential. Sumayya et al. [27] explored the stabilisation of expansive soil treated with tile waste, demonstrating the reduced swell potential and improved soil strength. Moreover, tile waste has been utilised in road pavement subgrade applications. Cabalar et al. [28] investigated the use of waste ceramic tiles for road pavement subgrades, presenting the potential of tile waste to improve the performance and durability of road subgrades. Rani et al. [29] investigated the strength behaviour of expansive soil treated with tile waste. The researchers evaluated the effectiveness of incorporating tile waste as a stabilising agent in improving the engineering properties of expansive soils. The results demonstrated notable improvements in the strength characteristics of the treated soil, highlighting the potential of tile waste to enhance the stability and load-bearing capacity of expansive soil.

Using ceramic dust from tile waste in soil stabilisation has been a subject of interest. Sabat [30] researched the stabilisation of expansive soil using ceramic waste dust, showcasing its effectiveness in improving the geotechnical properties of the soil. Al-Bared et al. [31] also suggested that the powder of tile waste with different microstructure sizes smaller than 63 µm should be studied to explore the stabilising effect. At least 66% of the natural pozzolan must fall through a 45 µm wet sieve to meet the requirements of ASTM C 618 [32]. There is no direct study investigating the effect of the size of the powdered tile waste, which is smaller or similar to traditional stabilisers such as cement or lime, on mechanical and microstructural properties. Therefore, the main objective of this study is to investigate the effect of smaller tile waste particles on soft soil. It is specifically aimed to explore the particle reaction and pozzolanic reaction by conducting tests on the plasticity index, compressibility, mechanical properties (unconfined compressive strength, indirect tensile strength, and flexural strength), and microstructural analysis with varying amounts of tile waste (0%, 5%, 10%, 15%, 20%, 30%, and 40%).

The use of recycled tile waste in construction will have a positive impact on the industry. Incorporating recycled tile waste into construction projects can promote innovation and development in the industry, which can create new products, techniques, and applications. This could result in a more sustainable and cost-effective construction approach while reducing waste and promoting environmental responsibility.

## 2. Materials and Methods

### 2.1. Materials

#### 2.1.1. Soil

In this study, the soil was excavated and collected from a site 1 m below the ground surface in Kuala Sungai Baru, Melaka, Malaysia. The high moisture content of the clay soil caused the dirt to clump into larger chunks. As a result, the first step in preparing the soil for testing was to physically separate the clay particles using a rubber mallet hammer until they were much finer, as shown in Figure 1. After that, the soil sample was placed in the oven to dry for 24 h, and then a disk mill grinder was used to decrease the size of the clay particles further. The properties of the soil are listed in Table 1. According to The Unified Soil Classification System (USCS), the soil is defined as clay with low plasticity (CL), where the soil lies above the A-line. The chemical composition and mineralogy of the soil were investigated using wavelength-dispersive X-ray fluorescence spectrometry (XRF). The results are listed in Table 2.

#### 2.1.2. Tile Waste

To evaluate the potential of tile waste as a stabiliser, tile waste powder was mixed with low-plasticity clays. The tile waste was collected from the construction site at the University of Malaya, Malaysia (Figure 2). The tile waste was crushed into smaller sizes using an aggregate crusher in the first step. Then, the tiles were ground using the Los Angeles Abrasion Machine. The grinding of the tile waste was carried out for 18 h, and the tile waste fineness was checked at 4.5 h grinding intervals using a 45 μm sieve. Based on the particle size distribution (Figure 3), the tile waste consisted of 7.5% sand friction, 70% silt size friction, and 22.5% clay friction. The tile waste appeared coarser than clay, as evidenced by the differences in their median sizes (D50), which equaled 8.57 μm for tile waste and 1.86 μm for clay. The specific gravity of the tile waste was 2.27, and the mass of tile waste passing the 45 μm sieve was 91%. The sieve fineness of the tile waste for different grinding periods is shown in Figure 4. According to Safiuddin et al. [33], the finer the grain size of the pozzolanic material, the greater the strength development. After the grinding process was completed, the particle properties of the tile waste were examined using a scanning electron microscope (SEM). The particles of the ground tile waste observed by SEM were angular and irregular, as seen in Figure 5. The particles had a rough surface, and the size ranged from small to large particles. The tile waste mainly consisted of SiO₂ and Al_2_O_3_, with both oxides accounting for 88.69% of the total mass. According to ASTM C 618, the tile waste was highly pozzolanic due to the cumulative mass content of silicon dioxide (SiO₂), aluminium (Al₂O₃), and iron oxide (Fe_2_O_3_), which was 91.06%. An X-ray fluorescence (XRF) analysis was performed to determine the main chemical composition of the tile waste, as shown in Table 2. In addition, the loss on ignition (L.O.I.) was 1.33, which fulfilled the requirement stated by ASTM C618 as a pozzolanic material.

### 2.2. Laboratory Works and Testing

#### 2.2.1. Sample Preparation

Previous research has shown that tile waste (of various sizes) can be utilised to stabilise soil by up to 30%. In this study, the amount of tile waste was added up to 40% to investigate the effect of high tile waste content in the soil stabilisation process. The content of tile waste used in this investigation had dry weight contents of 5, 10, 15, 20, 30, and 40%. At first, the soil was oven-dried for 24 h and then ground until it passed through a 2 mm sieve. The sieved soil was then mixed with specified tile waste content, as listed in Table 3. Water was then added to the dried mixture to achieve the estimated water content. The mixture was then compacted into three similar layers. Each layer was compacted at 36 strokes according to the optimum moisture content (OMC) and maximum dry density (MDD) determined during the compaction test. The soil samples were prepared using a mini compaction apparatus to perform an unconfined compressive strength test (UCS), indirect tensile strength test (ITS), and flexural strength test (FS). After compaction, the specimens were extruded from the mould, and wrapped in thin plastic film, and then in aluminium foil to prevent moisture loss. The specimens were sealed in an airtight Ziplock bag before being stored in a polystyrene box and cured for 7, 28, and 90 days.

#### 2.2.2. Atterberg Limits Test

The Atterberg limits were determined according to ASTM D4318 [34]. The liquid and plastic limits were tested on untreated and treated soil samples. For each test of the liquid and plastic limits, sufficient material from the portion of the soil (Table 3) that passed the 425 µm sieve was mixed with water to form a homogeneous paste.

#### 2.2.3. Compaction Test

Mini-compaction tests were conducted on untreated and treated samples with tile waste ranging from 0 to 40% to determine the maximum dry density and optimum moisture content using an apparatus developed by Sridharan and Puvvadi [35]. The mixes were compacted in a cylindrical mould (50 mm diameter and 100 mm height) at 36 strokes/layer in three layers using a 45 mm diameter rod. The proposed apparatus is simpler, faster, requires less labour, and saves much soil [35]. Loi et al. [36] also recommended this mini-compactor as a quick test to estimate the compaction behaviour in their work by comparing standard Proctor compaction tests and mini-compaction tests.

#### 2.2.4. Unconfined Compression Test

The unconfined compressive strength for the untreated and treated stabilised specimens was determined using the Instron machine. The test was carried out at 7, 28, and 90 days. The test procedure was used to determine the approximate strength of the soil samples according to ASTM D2166-13 [37]. These tests were performed on cylindrical samples with a height of 100 mm and a diameter of 50 mm. The specimens were loaded to failure at a 1.0 mm/min rate. The unconfined compressive strength was determined as the maximum load per unit area, and the total stress–strain curve was obtained from each test.

#### 2.2.5. Indirect Tensile Strength Test

The tensile strength of soil can be tested by an indirect method. The indirect method allows the development of correlations between different parameters to determine the tensile strength of the soil. Indirect tensile strength (ITS) was conducted according to the indirect Brazilian test [38] with cylindrical specimens of 50 mm diameter × 100 mm length. The samples were tested at 7, 28, and 90 days. The sample was positioned in the centre fixture and the specimens were split in the test under a linear compressive load. The compressive force continuously increased at a 1.0 mm/min rate until failure occurred along this plane. The ITS test was calculated by using Equation (1) below:ITS = 2Pult/πtD (1)
where ‘Pult’ is the maximum vertical load applied in the test (N), ‘D’ is the diameter of the sample (mm), and ‘t’ is its thickness or height (mm).

#### 2.2.6. Flexural Strength Test

The flexural strength test was carried out on a 50 mm diameter and 100 mm cylindrical soil mixture at 7, 28, and 90 days. The three-point bending test with a supported span at a length of 60 mm was carried out according to the modified test procedures of Vivi Anggraini [39] with reference to ASTM D 1635 [40]. The load was then applied uniformly and continuously at a rate of 0.1 mm/min until failure. The flexural tensile strength can be calculated by using Equation (2) below:σf = PL/πr^3^
(2)
where ‘P’ is the failure load (N), ‘L’ is the distance between the support rollers (mm), and ‘r’ is the radius of the sample (mm).

#### 2.2.7. Microstructure and Mineralogy Analysis

The surface of both untreated and treated samples was examined using scanning electron microscopy (SEM, Phenom XL Desktop, Thermo Fisher Scientific, Waltham, MA, USA) to observe the pozzolanic reactivity and visualise crystallised compound formation. An oven-dry sample of less than 10 mm from the UCS test was mounted on an aluminium stub using carbon tape. SEM imaging was taken at 15 KV at 20 μm scale or at 3000× magnification. Additionally, energy-dispersive X-ray spectrometry (EDS, OmniProbe 200, Oxford Instrument, Abingdon, UK) was used to investigate the elemental compositions on the surface of the analysed samples during imaging. Meanwhile, X-ray diffraction (XRD) was used for examining the mineralogy that occurred in the sample before and after the stabilisation process. A similar sample used in SEM was pulverised and scanned using a PANalytical X-ray diffraction spectrometer. The samples were scanned at a 2Ɵ angle with a range of 6° to 90° at 0.02° per step. Through the analysis of variations in the diffraction pattern across different samples, it was possible to detect the shifts in mineralogical composition and gain a better understanding of the underlying physical and chemical processes that have taken place.

## 3. Results

### 3.1. Plasticity

This study investigated the changes in the liquid limits, plastic limits, and plasticity index due to the addition of tile waste to the soil. Figure 6 shows the results of the liquid limit, plastic limit, and plasticity index. The liquid limit and plasticity index decreased as the percentage of tile waste increased. It is seen that the value of the liquid limit decreased from 48.5% to 33.5%. Adding 5% of the tile waste to the original soil was found to increase the plasticity limit. However, the plasticity limit value was decreased by incrementing 10% tile waste content, which continued until 40% tile waste content was added.

The plastic limit is the water content at which soil starts to exhibit plastic behaviour. It is the minimum water content at which the soil can be rolled into a thread of 3 mm diameter without breaking. The plastic limit of soil indicates its ability to withstand deformation without cracking. Soils with higher plastic limits are more resistant to deformation and are considered more stable. In the given table, the plastic limit values ranged from 26.8% to 24.0%, indicating an increase in soil stability as the plastic limit increased.

The soil mixture required less water to change from the plastic state to the semi-solid state when tile waste was present. The decrease in the plasticity index observed in the samples was due to the agglomeration and coagulation of clay induced by the addition of tile waste [22]. From the plasticity chart (Figure 7), the addition of ceramic tile waste changed the soil from inorganic clay with low to medium plasticity (CL) to inorganic silt with low plasticity (ML). The shift in soil class was due to the addition of coarse-grained particles from tile waste to the fine-grained particles of clay soil. The same pattern was also reported by [28,30,41].

### 3.2. Compaction Test

The relationship between OMC and MDD for different percentages of tile waste is shown in Figure 8. The optimum moisture content (OMC) decreased with an increasing percentage of tile waste. The OMC decreased sharply when 0% to 15% of tile waste was added to the mix. The decrement trend continued until 40% of tile waste addition. This is because, with the addition of tile waste powder, the water-holding capacity within the stabilised clay particles decreased, leading to a decrease in OMC [29]. Sabat [30] also stated that replacing ceramic dust particles with soil particles reduces the attraction for water molecules, hence decreasing the optimum moisture content (OMC).

Meanwhile, the maximum dry density values followed a reverse trend. Increasing the proportion of tile waste led to an increase in MDD, indicating an increase in the density of the sample. This correlation is likely because the sample became more compact and denser as the moisture content decreased, resulting in a higher MDD. The admixture of tile waste grains with higher specific gravity and soil grains with low specific gravity increased the MDD. These results are in agreement with the findings of other researchers [25,29,30,42]. Therefore, adding tile waste with a different specific gravity improves the compatibility of the untreated soil.

### 3.3. Effect on the Unconfined Compression Strength (UCS)

Seven different soil mixtures, labelled as S (untreated soil), ATW5, ATW10, ATW15, ATW20, ATW30, and ATW40, were prepared in this study. UCS tests were conducted on these soil mixtures at three different curing times, namely, 7 days, 28 days, and 90 days. Figure 9 shows the results of compressive stress versus compressive strain for different proportions of tile waste, ranging from 0% to 40%. The maximum UCS values were extracted based on this graph (Figure 9). The percentage improvement of the treated soil sample was obtained by calculating the percentage increase in strength achieved by a treated soil sample compared to a (control) untreated sample.

The results in Table 4 show that the UCS values of all soil mixtures increased with the curing time. The ATW15 mixture indicated the highest UCS values at all the curing times, followed by ATW10 and ATW5. Meanwhile, the UCS values of the untreated soil (S) were the lowest among all mixtures at all curing times. Additionally, it was observed that the UCS values for some mixtures, such as ATW5 and ATW15, increased significantly from 7 days to 28 days. However, there was a slight increment in the UCS values from 28 days to 90 days. In contrast, the UCS values of other mixtures, such as ATW10 and ATW20, increased significantly from 28 days to 90 days. This indicated that the curing time significantly impacted the soil strength. As the curing time increased, the soil mixtures gained strength. In addition, the rate of strength varied for different soil mixtures, and it was also found that some of the mixtures achieved higher strength more rapidly than others.

From Figure 10, it can be concluded that the addition of tile waste improved the UCS value of the untreated soil. As per the analysis, the optimum content for the tile waste addition was 15%, as in Table 4. This is because the soil mixture with 15% tile waste yielded the highest UCS value, namely, 0.674, 0.710, and 0.786 MPa, with a strength improvement of 27.45, 27.89, and 26.97% at 7, 28, and 90 days of curing time.

The UCS of the treated soil increased by the tile waste content until the optimum tile waste mixture was reached. Any additional tile waste content beyond the optimum value showed a decrement in UCS. The decrease in strength, especially for the treated soil mixtures, was lower than the untreated soil, which could be due to excessive tile waste content in the soil without or with less formation of the pozzolanic reaction. The presence of tile waste in the mixture acted as a filler, and less cohesion was formed between the particles due to the high proportion of tile waste.

Elsawy [42] observed a similar behaviour pattern for the performance of dune sand in stabilising highly expansive soils, and Kollaros and Athanasopoulou [43] for sand as a soil stabiliser. The increase in the strength of samples was due to modification and stabilisation. The ion exchange reaction is called modification, and the formation of a pozzolanic reaction is called stabilisation, which reduces the porosity of the treated soil [25]. With the addition of tile waste, it was indicated that the strength of the treated soil not only depended on the particle interaction between soil and tile waste but also on the pozzolanic reaction between particles. This reaction provided an additional binding between soil particles and tile waste particles, contributing to its strength.

### 3.4. Effect on the Indirect Tensile Strength Test (ITS)

It was indicated in Figure 11 that the ITS values of the mixtures exhibited significant increments with curing time, except for ATW 30 and ATW 40, which indicated a slight increment in ITS values as compared to other mixtures. In addition, at 7 days, the ITS values ranged from −55.98% lower for ATW 40 to 27.77% higher for ATW 15 compared to the untreated soil sample. By 28 days, the ITS values had generally increased for all of the mixtures except for ATW 30 and ATW 40, which showed a lower strength than the untreated soil sample. At 90 days, the ITS values continued to increase for most of the mixtures, with ATW 15 showing the highest percentage increase of 76.57% compared to the untreated soil sample. This showed that adding 15% tile waste indicated an optimum result for the ITS performance. However, ATW 30 and ATW 40 showed a decreased strength of −7.12% and −35.66%, respectively, compared to the untreated soil sample. Any addition of tile waste beyond the optimum will result in a strength drop. The weak mechanism interaction between the tile waste and the soil particles, as well as little or no pozzolanic reaction, led to a decrement in the tensile strength.

The observed trend in the ITS values of the mixtures was expected as the strength gained over time and as the cementitious material in the mix hydrated and bound the materials together due to the presence of magnesium and sodium, which acted as catalysts in the formation of specific cementitious compounds [26].

Figure 12 illustrates the relationship between the indirect tensile strength and the unconfined compressive strength at 28 days when the tile waste was added to the mixture. In this study, the indirect tensile strength varied between 32.42 kPa and 81.67 kPa, while the unconfined compressive strength ranged between 465 kPa and 778 kPa. They varied equally depending on the content of the tile waste. Thus, a significant positive linear relationship was found between the indirect tensile strength and the unconfined compressive strength, with a correlation coefficient (R^2^) of +0.9664 (Figure 12). A similar correlation was observed by Solanki and Zaman [44] for the subgrade soil with the incorporation of lime, fly ash, and cement kiln dust.

### 3.5. Effect on the Flexural Strength Test (FS)

Figure 13 shows the flexural load versus the displacement of flexural strength for different tile waste content ranging from 0% to 40%. Similar to the UCS and IDT results, the optimum content of the tile waste addition was 15%, a significant increase in the peak flexural load, which was 0.206 MPa (43.45%), 0.243 MPa (52.17%), and 0.299 MPa (61.07%), of flexural strength for 7, 28, and 90 days of curing.

The FS of the treated soil increased by an increment of tile waste contents until the optimum content had been reached. Any additional tile waste content beyond the optimum amount indicated a decrement in FS values. In general, results from the flexural test were consistent with IDT and UCS. This is primarily attributed to the reaction between clay and tile waste that generated cementitious compounds by improving the cohesion between soil particles, thereby augmenting the soil’s tensile strength.

However, it should be noted that the FS strength did not solely rely on the presence of cementing agents that connected soil particles through the formation of cementitious compounds. It also depended on the availability of adequate materials (such as tile waste particles) to fill the voids within the soil. Consequently, the impact of tile waste on flexural strength was greater than on UCS.

The relationship between the unconfined compressive strength and flexural strength was analyzed with the linear regression and correlation analysis as presented in Figure 14. Through the analysis, it was indicated that the flexural strength had a strong correlation with R^2^ = 0.9024 for ATW.

### 3.6. Effect on Microstructures

The SEM images of the untreated compacted clay samples are presented in Figure 15a. At the optimum moisture content, the compacted sample is likely to develop a dispersed structure due to the presence of sufficient pore water that creates a complete double layer of ions attracted to the clay particles. Therefore, the clay particles and bands can easily slide over each other during shearing, leading to lower strength and stiffness [45].

On the contrary, Figure 15b shows SEM micrographs of ATW with 15% TW cured for 28 days. The presence of crystallised compounds (white lump) can be observed on the surface and between the particles due to the reaction between the soil particles and tile waste. Non-reacting particles bound together with the formation of a white lump created a denser microstructure that contributed to the strength of the soil. Al-bared et al. [23] reported similar observations in their findings, which found that the new crystallised compounds created less porous and denser samples. Figure 15b shows the physical interaction between the soil, tile waste particle, and crystallised compounds. These bounding mechanisms directly influenced the strength of the soil because of the interfacial zone and formation of a cohesive structure.

Figure 16 shows the EDS analysis of the (a) untreated clay sample and (b) treated clay with 15% tile waste cured for 28 days. The major elements of the untreated clay were Si, O, Al, Fe, and K. Meanwhile, the tile waste-stabilised soil sample consisted of SI, Al, K, Mg, and Ca. The analysis shows that the new element (Mg and Ca) detected in the stabilised sample was responsible for the formation of magnesium silicate hydrate (M-S-H) and calcium-alumina-silicate-hydrates (C-A-S-H). Furthermore, it can be inferred that utilising tile waste is a feasible method for imparting greater shear strength and stiffness to clayey soils. This is because the similarities between the conditions of using tile waste and those of granular soils are expected to substantially influence the internal friction angle parameter of the original soil. Therefore, tile waste can treat soils as a filler and a binder that chemically reacts with the soil components [31].

### 3.7. Effect on Mineralogy

X-ray diffraction (XRD) testing is commonly employed to identify phases in crystal material samples. Figure 17 depicts the XRD patterns for untreated soil and ATW15 samples undergoing a 7-day curing process. Table 5 shows the compound identification for untreated soil and treated soil with 15% TW. The compounds identified in each sample are listed along with their respective chemical formulas or elemental compositions and semi-quantitative (S-Q) percentages. Based on this XRD analysis, Quartz, Illite, and Kaolinite were the dominant minerals identified in the untreated soil. However, the addition of tile waste in the untreated soil influenced the quantity of the mineral (Quartz), and some minerals (Illite and Kaolinite) were not detected in the treated soil, which indicates that these minerals had reacted with the tile waste to become new compounds. The S-Q of Quartz decreased from 45% in the untreated soil sample to 29.3% in the treated soil sample. The reduced quantity of Quartz can be attributed to the partial destruction of silicate minerals during the pozzolanic reaction between soil and tile waste. James and Pandian [22] reported that the addition of TW expedited the pozzolanic reactions that resulted in the disintegration of the mineral crystal structure. The reduction also increased the formation of new compounds in the treated soil sample.

The treated sample showed a formation of Bavenite at 25.34°, 35.12°, 58.95°, and Brucite at 19.95° as a new compound. The Bevenite formation associated with C-A-S-H calcium-based hydration compounds is caused by the hydration reaction of calcium with the alkalis in the mixture [46]. The physicochemical properties of the cementitious structure are enhanced when Bavenite develops in the presence of calcium silicate hydrate [47]. The EDX analysis confirmed the presence of Ca and Mg elements, consistent with the mineralogical composition observed through XRD (C-A-S-H and M-S-H). The XRD revealed structures, phases, and crystal orientations, while EDS with SEM provided qualitative and semi-quantitative data [48]. Unluer and Al-Tabbaa [49] reported that Brucite could fill up the available pores and increase the density of the soil due to the volume expansions associated with the formation of Brucite. Magnesium oxide (MgO) initially hydrates to Mg(OH)₂ (Brucite) during this process. A carbonate phase is formed when CO₂ is absorbed by the hydrated material and mixed with magnesium oxide and water. When exposed to air, various cementitious materials undergo carbonation reactions with the carbon dioxide present in the atmosphere [50]. Mg enrichment is typically linked to the precipitation of Mg(OH)₂ (Brucite).

The XRD revealed changes in mineral composition due to the addition of tile waste, with a reduction in Quartz intensity, and the formation of Bavenite and Brucite crystals. These reactions contribute to the densification of the sample, which can fill up the available pore space in the soil and increase its density.

## 4. Conclusions

Based on the experimental results for the soil stabilisation with different percentages of tile waste produced in the study, the following conclusions can be drawn:The changes in the liquid limits, plastic limits, and plasticity index were due to the addition of tile waste to the soil. The results revealed that as the percentage of tile waste increased, the liquid limit and plasticity index decreased. Adding 5% of the tile waste to the original soil was found to increase the plasticity limit. However, the plasticity limit value was decreased by incrementing 10% tile waste content until 40% tile waste content was added. The plastic limit values indicated an improvement in soil stability as they increased. The presence of tile waste reduced the water content required for the soil to transition from plastic to a semi-solid state. As supported by previous studies, the decrease in the plasticity index was attributed to clay particle agglomeration induced by the tile waste addition. Referring to the USCS chart, the soil group changed from CL to ML with the presence of tile waste in the soil sample.With the addition of tile waste, the water-holding capacity inside the stabilised soil particles decreased, resulting in a decrement in OMC, with a significant decrement occurring in between 0% and 15% of the tile waste content. This trend continued until the 40% tile waste addition. Conversely, the increment in the tile waste percentage led to increasing MDD, indicating an increment in the sample density. The combination of tile waste grains with higher specific gravity and soil grains with lower specific gravity contributed to the higher MDD values. A uniform gradation of particles in the soil was created with tile waste in the mixture. This will decrease the void ratio and make the soil more compact.The mechanical strength for UCS, ITS, and FS improved when the optimum percentage of tile waste (15%) was added. The addition of 15% tile waste indicated the highest UCS values and strength improvements of 27.45%, 27.89%, and 26.97% at 7, 28, and 90 days of curing time, respectively. Meanwhile, the ITS and FS values also increased with the addition of the optimum percentage of tile waste. Any addition beyond the optimum tile waste content indicated a decrement in strength due to excessive tile waste content in the soil samples without a pozzolanic reaction.The SEM and XRD data analysis for ATW15 showed that a crystalline compound (Brucite and Bavenite) had formed on the soil sample’s surface, which played a crucial role in mechanical strength development. The similarities between tile waste and granular soils can significantly impact the internal friction angle of the original soil, treating it as a filler and binder that chemically reacts with soil components. The physical interaction between soil, tile waste particles, and crystallised compounds plays a vital role in this improvement. These interactions create cohesive structures and affect the strength of the soil through the interfacial zone.

## Figures and Tables

**Figure 1 materials-16-05261-f001:**
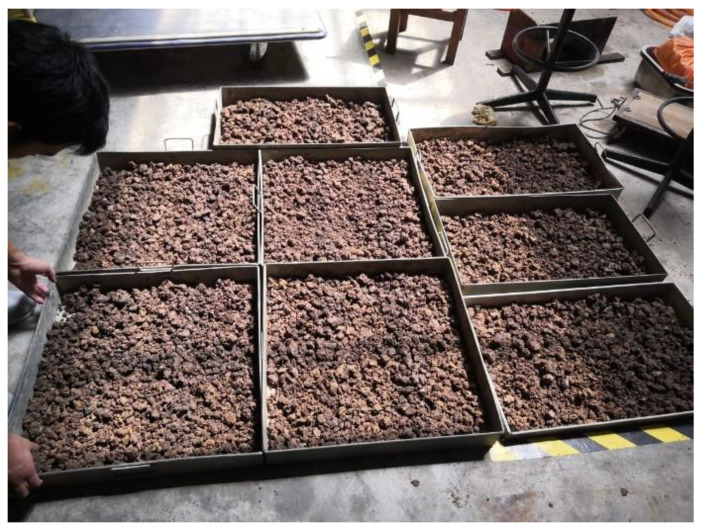
Clay soil sample.

**Figure 2 materials-16-05261-f002:**
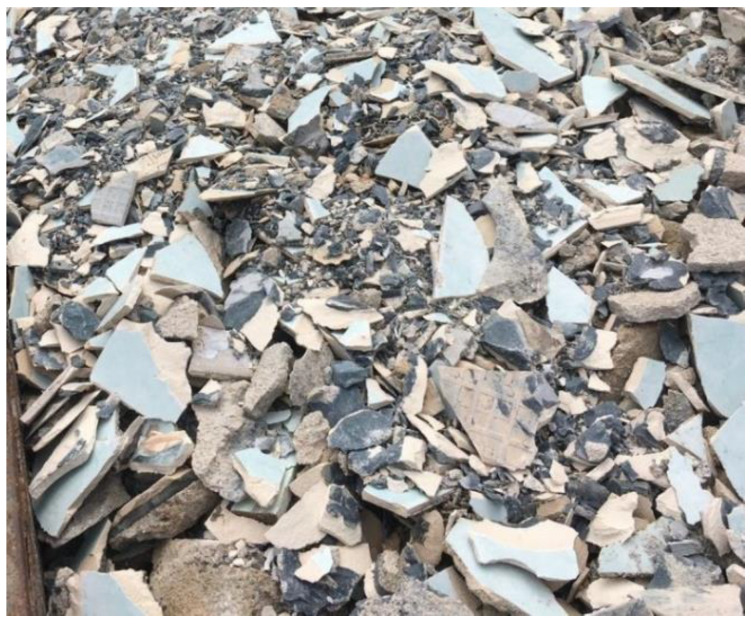
Tile waste.

**Figure 3 materials-16-05261-f003:**
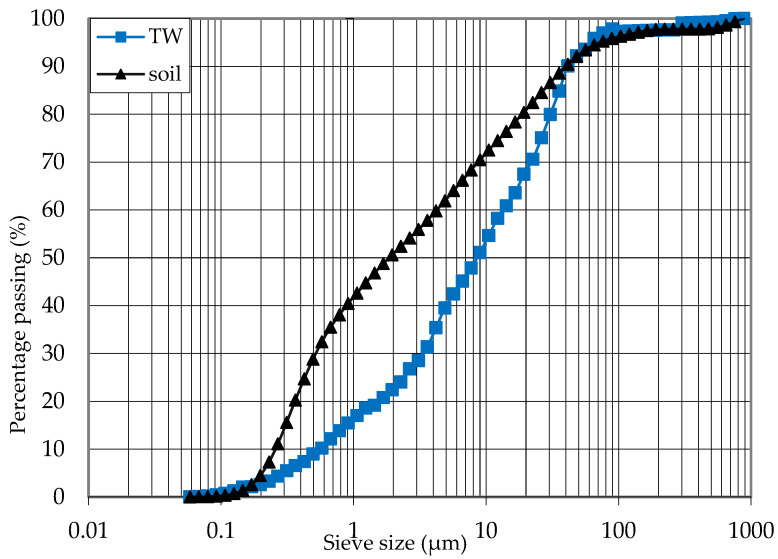
Particle size distribution of soil and ground tile waste.

**Figure 4 materials-16-05261-f004:**
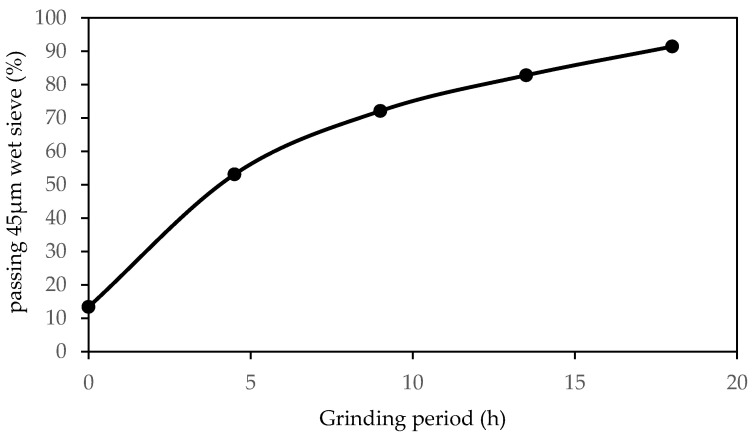
The fineness of tile waste with different grinding periods.

**Figure 5 materials-16-05261-f005:**
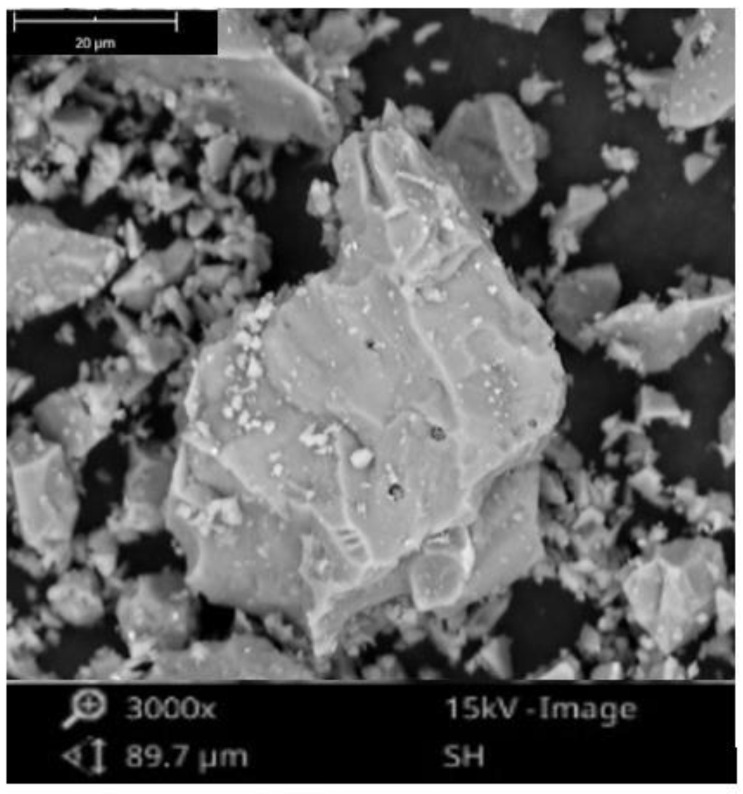
Scanning electron micrographs (SEM) of ground tile waste.

**Figure 6 materials-16-05261-f006:**
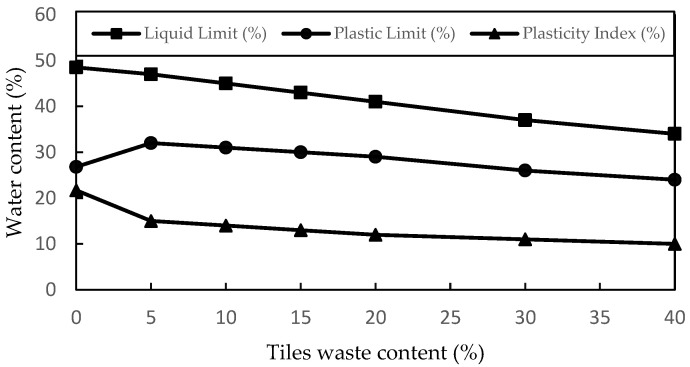
Atterberg limit of soil with the addition of tile waste.

**Figure 7 materials-16-05261-f007:**
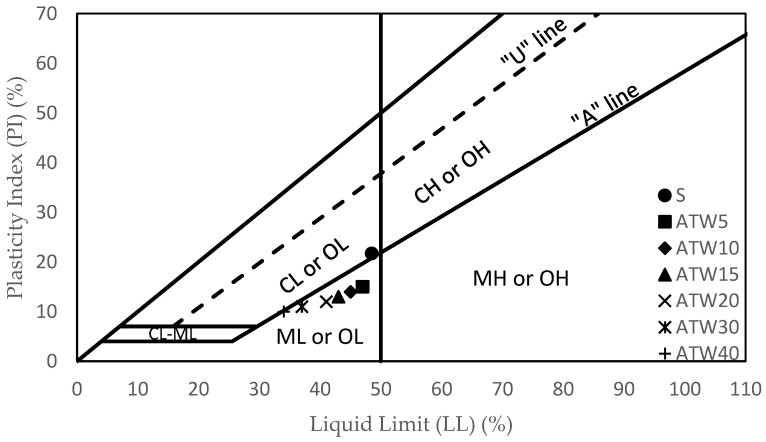
(Unified Soil Classification System) plasticity chart indicating the position of soil and soil with tile waste.

**Figure 8 materials-16-05261-f008:**
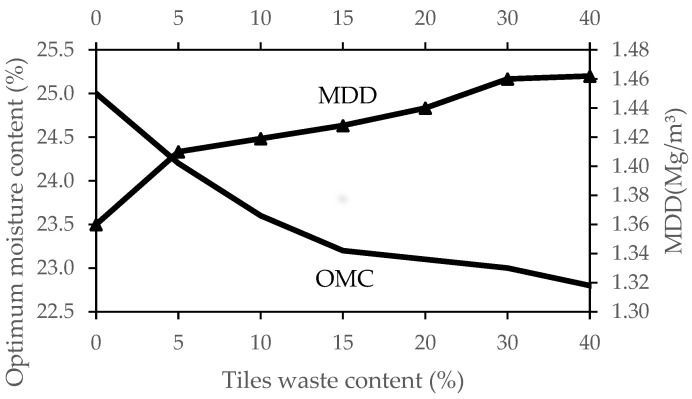
OMC and MDD of soil for different tile waste content.

**Figure 9 materials-16-05261-f009:**
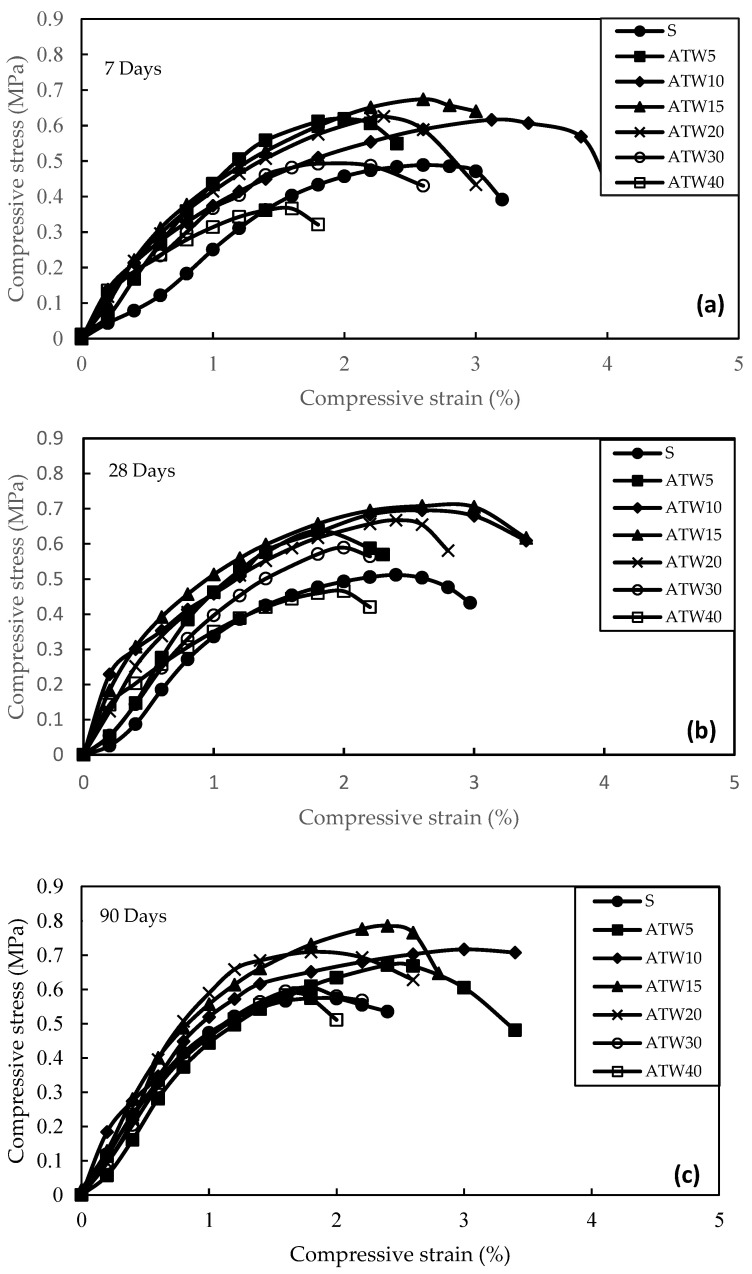
Compressive stress vs. compressive strain for different mixtures at (**a**) 7 days, (**b**) 28 days, and (**c**) 90 days.

**Figure 10 materials-16-05261-f010:**
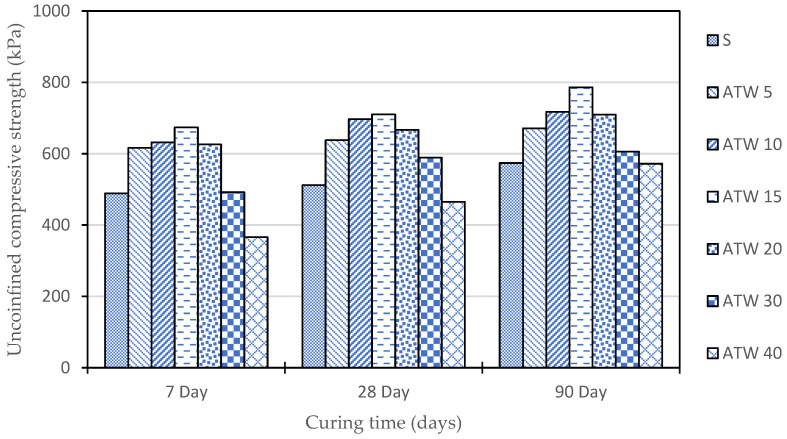
UCS results of treated tile waste.

**Figure 11 materials-16-05261-f011:**
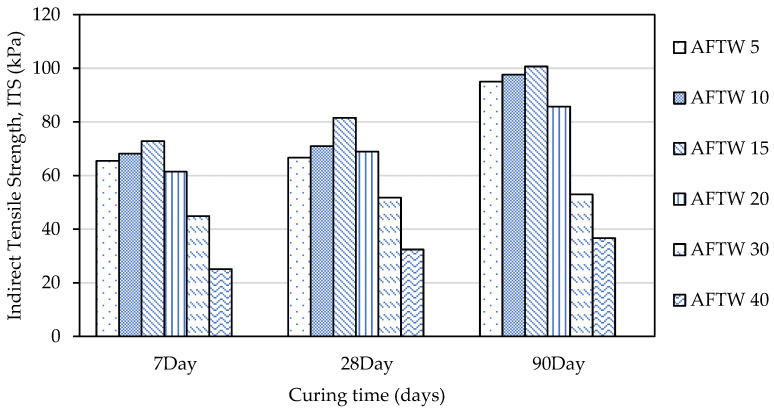
Indirect tensile strength of ATW sample.

**Figure 12 materials-16-05261-f012:**
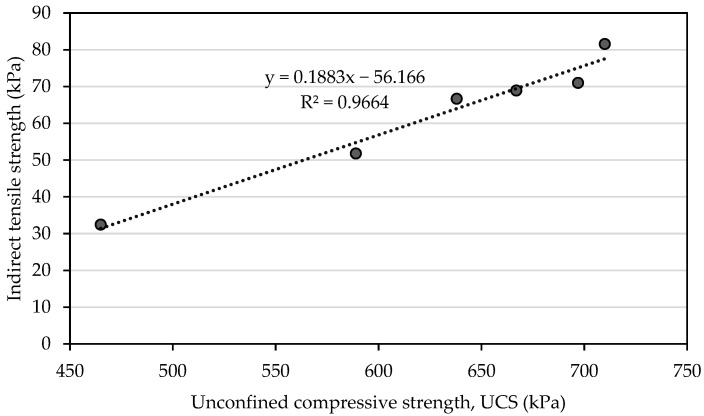
Correlation between indirect tensile strength test and unconfined compressive strength test for 28 days.

**Figure 13 materials-16-05261-f013:**
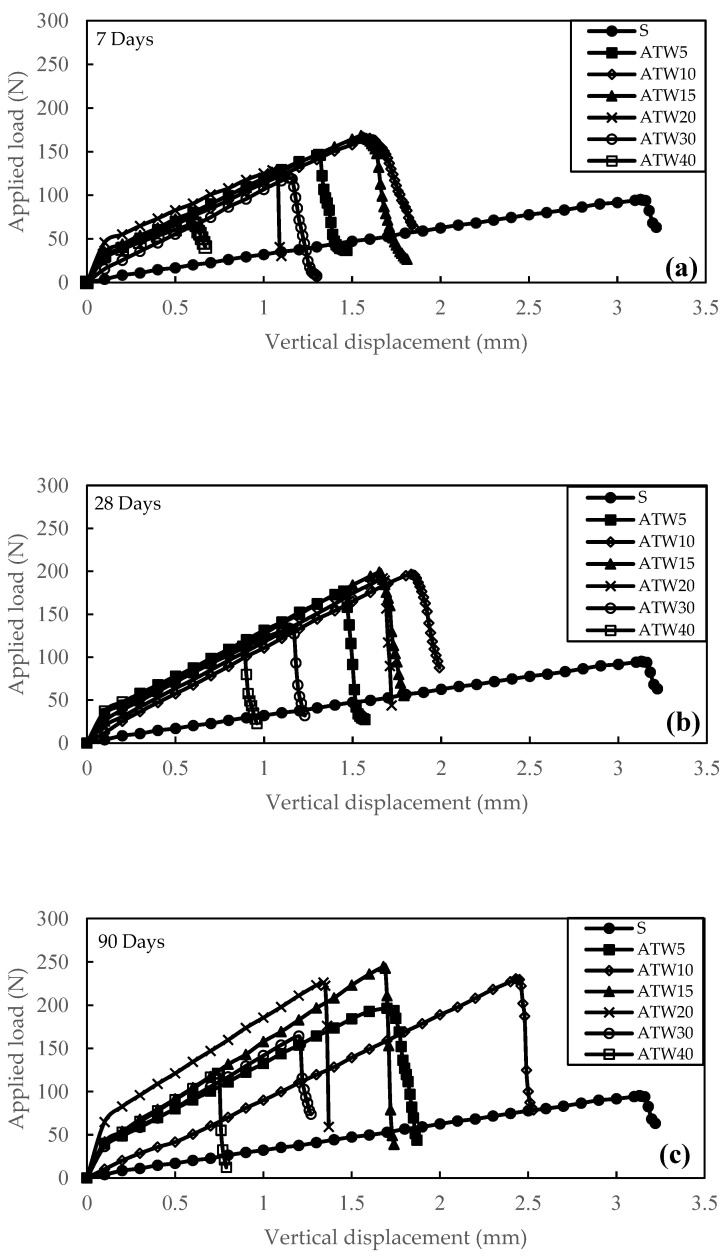
ATW flexural load versus displacement for (**a**) 7 days, (**b**) 28 days, and (**c**) 90 days.

**Figure 14 materials-16-05261-f014:**
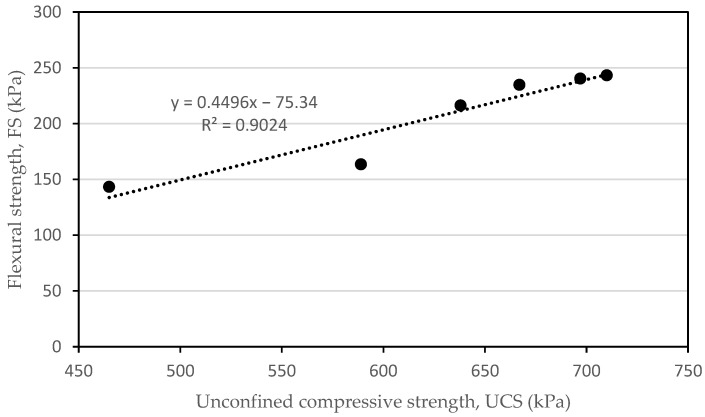
Correlation between flexural strength test and unconfined compressive strength test for 28 days.

**Figure 15 materials-16-05261-f015:**
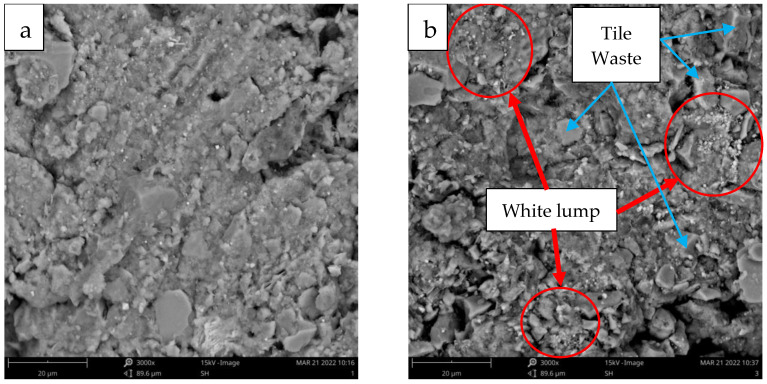
SEM micrographs of (**a**) untreated clay sample and (**b**) treated clay with 15% tile waste cured for 28 days.

**Figure 16 materials-16-05261-f016:**
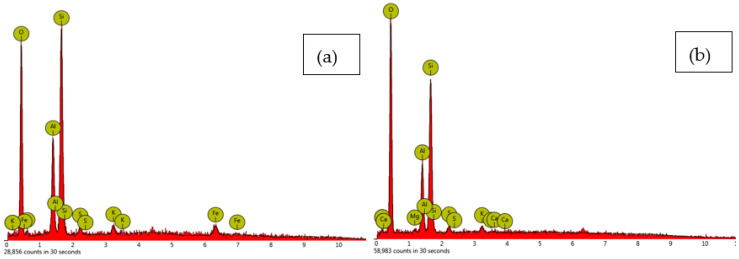
The EDX spectrum of (**a**) untreated clay sample and (**b**) treated clay with 15% tile waste cured for 28 days.

**Figure 17 materials-16-05261-f017:**
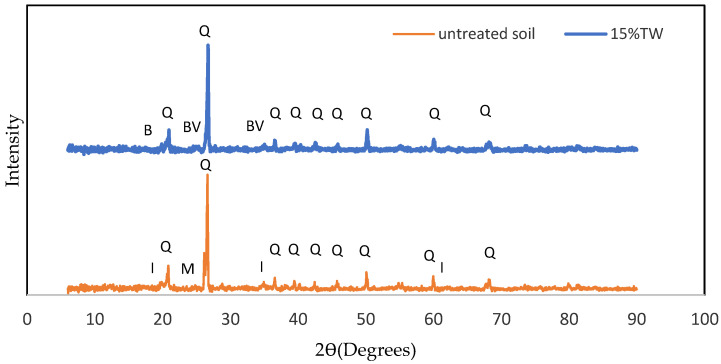
XRD scattering graph for untreated soil and 15% TW (Q: Quartz, I: Illite, M: Montmorillonite, BV: Bavenite, and B: Brucite.

**Table 1 materials-16-05261-t001:** The properties of the soil.

Properties	Values
Specific gravity, Gs	2.50
Atterberg limits	
Liquid limit, LL (%)	52
Plastic limit, PL (%)	22
Plasticity index, PI (%)	30
Mechanical properties	
Optimum moisture content, OMC (%)	26
Maximum dry density, MDD (Mg/m^3^)	1.46
Unconfined compressive strength, UCS (kPa)	557

**Table 2 materials-16-05261-t002:** Major chemical composition of soil and tile waste used in this study.

Chemical Composition	Content (%)	
	Soil	Tile Waste
SiO_2_	64.70	71.68
CaO	0.02	1.10
Fe_2_O_3_	6.91	2.38
Al_2_O_3_	17.83	17.00
Na_2_O	0.13	3.45
SO_3_	6.85	0.03
MgO	0.72	1.00
K_2_O	1.34	2.28
TiO_2_	1.10	0.38
LOI (loss on ignition)	10.8	1.33

**Table 3 materials-16-05261-t003:** Mixture proportions of the various tests.

Group Series	Sample Code	Samples	Curing Days
S	S	Natural Soil	7, 28, & 90
ATW	ATW5	soil + 5% Tile Waste	7, 28, & 90
	ATW10	soil + 10% Tile Waste	7, 28, & 90
	ATW15	soil + 15% Tile Waste	7, 28, & 90
	ATW20	soil + 20% Tile Waste	7, 28, & 90
	ATW30	soil + 30% Tile Waste	7, 28, & 90
	ATW40	soil + 40% Tile Waste	7, 28, & 90

Where S = natural soil (untreated/control); ATW = additional tile waste.

**Table 4 materials-16-05261-t004:** Effect of curing time on UCS values of various soil–tile waste mixtures.

Mixture	UCS Value at the Curing Time, kPa		
	7 Days	28 Days	90 Days
S (untreated soil)	489	512	574
ATW5	616	638	671
ATW10	632	697	717
ATW15 (optimum)	674	710	786
ATW20	626	667	710
ATW30	492	589	606
ATW40	366	465	572

**Table 5 materials-16-05261-t005:** Compound identification for untreated soil and treated soil with 15% TW.

Mixture	Compound	Formula/Elements	S-Q (%)
S (untreated soil)	Quartz	Si3.00O6.00	45
	Illite	K2.00Al4.00Si8.00O24.00	48
	Kaolinite	Al8.00Si8.00O36.00	7
ATW15 (optimum)	Quartz	Si3.00O6.00	29
	Brucite	Mg1.00O2.00H2.00	3
	Bavenite	Ca16.00Si36.00Be12.00Al4.00O112.00H12.00	67

## Data Availability

All the data have been included in this paper.

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
