# Peer review of "Structural Characteristics and Microstructure Analysis of Soft Soil Stabilised with Fine Ground Tile Waste"

_materials, 2023, doi:10.3390/ma16155261_

Round 1

Reviewer 1 Report

What is the location of the site of excavation?

What are the Cu, Cc, and D50 values of the soil samples?

What are the D50 and LOI of the tile waste?

What are the settings used for SEM imaging?

What is the sample preparation method used?

What are the settings and sample preparation methods for XRD?

The correlation between the microanalyses and the strength observations are not technically informative.

The influence of adding tile waste on the microstructure of the soil, especially the pore characteristics and the interfacial zones have significant impact on the mechanical properties as well as the permeability.

This is not explained in the manuscript.

English language is fine but the authors are suggested to avoid using inanimate objects in the first person.

Such as "soil's strength" should be rewritten as "strength of the soil".

Reviewer 2 Report

The manuscript needs minor corrections. Find the comments attached, 

Reviewer 3 Report

The paper "Structural Characteristics and Microstructure Analysis of Soft Soil Stabilised with Fine Ground Tile Waste" needs improvement in language and content.

The English language should be revised.

The Introduction needs more investigations on the state of the art. Most of the sentences is not referenced. "destroy the integrity of the subgrade" at line 37 demonstrates the uncorrect use of the English language.

The subgrade is not designed to distribute the loads because the pavement has this role "subgrade must have sufficient strength and stiffness to distribute the loads".

The content at lines 57-63 should be referenced.

Figure 1, Table 1 and Table 2 should be moved near the text that cites them. The same for TAble 3.

Equations should be cited in the manuscript before they appear.

Line 375: if R2 is 0.59 as in Figure 12, the positive linear relation-
ship between the indirect tensile strength and the unconfined compressive strength is not significant positive (lines 373-374).

The legend in Figure 14 is not clear.

The conclusion does not present numerical results and should be rewritten.

The English language should be revised.

Reviewer 4 Report

The paper materials-2446755 "Structural Characteristics and Microstructure Analysis of Soft Soil Stabilised with Fine Ground Tile Waste" is interesting and within the scope of the journal.

I have read this article with great care and attention where I have made some comments/questions through which I will try to increase the importance and value of this valuable paper and this is the primary objective of evaluating any research articles.

The present manuscript deals with a combined XRD and SEM/EDS analysis of clay soils modified with recycled tile waste for different periods of time. Although this topic is both of fundamental and applied interest, I fear it is unsuited for publication in its present state for the following reasons:

 1) the experimental part lacks of important information missing, i.e. details about the used X-ray optics (e.g. filters, monochromators, type of detectors, soller slits, ...), and for the EDS analysis, no information at all is provided, in particular for the photon detection system (lower limit of photon energy, quantification, software, ...). This needs to be completely reworked.

2) the analysis of the XRD is inadequate and far to superficial for a scientific journal - it is just touching the surface of a data evaluation. I would recommend to check recommendations of the International Union of Crystallography, or related textbooks, or review articles. I furthermore feel that much more elaborate analysis has been performed by other groups in the past, i.e. carefully check the existing literature. I suggest considering:

Moretti, L., Natali, S., Tiberi, A., & D'Andrea, A. (2020). Proposal for a methodology based on xrd and sem-eds to monitor effects of lime-treatment on clayey soils. Applied Sciences (Switzerland), 10(7)

3) Some sentences should be simplified for readers to understand better.

I recommend that the authors take into account the comments listed above, as well as those of other reviewers (if any) in order to improve it before its reconsideration. The study is clear and adequately described. In my opinion Once the comments are made, the authors can resubmit the paper again for evaluation and possible publication in Materials.

Moderate editing of English language required

Round 2

Reviewer 1 Report

The reduction in intensity is not necessarily correlated with the quantity of a particular crystal in the whole specimen. The intensity is merely an indicator of the presence of near-perfect crystals (sharp peaks with high intensity if the x-rays are diffracted from the surface of that particular crystal) or semi-crystalline (smaller peaks with much lesser intensities) or amorphous ( only little humps and no sharp peaks). For estimating the quantities or degree of destruction of particular crystals, QXRD is suggested.

LOI is loss 'on' ignition, not 'of'.

Reviewer 3 Report

not excellent paper. It can be accepted

Author Response

Thank you for your comment.

Reviewer 4 Report

the paper can be accepted

Author Response

Thank you very much for your feedback.